# Pedestrian Evacuation Risk Assessment of Subway Station under Large-Scale Sport Activity

**DOI:** 10.3390/ijerph17113844

**Published:** 2020-05-28

**Authors:** Zeyang Cheng, Jian Lu, Yi Zhao

**Affiliations:** 1Jiangsu Key Laboratory of Urban ITS, Southeast University, Nanjing 211189, China; chengzeyang@seu.edu.cn; 2Jiangsu Province Collaborative Innovation Center of Modern Urban Traffic Technologies, Southeast University, Nanjing 211189, China; 3School of Transportation, Southeast University, Nanjing 211189, China; 4College of Automobile and Traffic Engineering, Nanjing Forestry University, Nanjing 210037, China; zhaoyi207@126.com

**Keywords:** pedestrian evacuation risk, pedestrian stampede probability, pedestrian casualty, subway station, large-scale sports activity

## Abstract

Pedestrian evacuation risk of subway stations is an important concern in city management, as it not only endangers public safety but also affects the efficiency of urban subway transportation. Determination of how to effectively evaluate the pedestrian evacuation risk of subway stations is of great significance to improve pedestrian safety. Previous studies about the pedestrian evacuation of subway station were primarily focused on pedestrian moving behaviors and the evacuation modeling, and the evacuation scenario is the regular subway operation. There is a dearth of studies to quantify the pedestrian evacuation risk in the evacuation process, especially the pedestrian evacuation risk quantitative characterization of subway station in large-scale sport activity. The current study develops a quantitative pedestrian evacuation risk assessment model that integrates pedestrian stampede probability and pedestrian casualty. Then several different simulation scenarios based on the social force model (SFM) are simulated to evaluate the pedestrian evacuation risk of the “Olympic Park Station” in Beijing, China. The results demonstrate that the pedestrian evacuation method, pedestrian stampede location, and distance from the stampede location to the ticket gate have a large impact on pedestrian evacuation risk. Then, the pedestrian evacuation scenarios with the lowest and highest risk for the “Olympic Park Station” in large-scale sport activity are determined. The findings have potential applications in pedestrian safety protection of subway station during large-scale sports activity.

## 1. Introduction

As an urban rail transit mode, subway transportation has some advantages of low pollution, convenient traffic, and low energy consumption, which is in conformity with the sustainable development of urban transport. However, sometimes, the subway station also presents some characteristics of high pedestrian density, closed space, and slow pedestrian speed, which decreases the pedestrian flow efficiency and the passengers’ transfer comfort, and even causes a pedestrian stampede. Especially during large-scale sports activities, the pedestrian flow of subway stations increases dramatically within a short time, which further increases pedestrian risk. Thereby, how to analyze the characteristics of pedestrian movement and evaluate the pedestrian evacuation risk plays a significant role in supporting the normal operation of subway station and ensuring pedestrian safety.

Numerous scholars have studied the influence factor of pedestrian evacuation of subway stations. Li and Han [1] simulated the pedestrian evacuation process using extended cellular automata model. The results illustrated that the pedestrian evacuation speed can slow down by both the highly conservative behavior and aggressive behavior of pedestrians. Chen et al. [2] proposed a pedestrian evacuation model of subway station based on Legion and Fires Dynamics Simulator (FDS) simulation under a fire event, and they established a dynamic evaluation indicator system based on survival index, security risk index, effectiveness index, and orderliness index. Wu et al. [3] estimated the evacuation capacity of subway stations in emergency conditions using the bilevel programming model. Some scholars also simulated the pedestrian evacuation in emergency condition, and they explored the relationship of evacuation time, subway station geometric features, and characteristics of passengers [4]. Lei et al. [5] explored the pedestrian evacuation process in different cases using an agent-based model. The relationship between occupant density, automatic fare gates, and evacuation time were analyzed. Yang et al. [6] investigated the pedestrian evacuation process under fire event, and further analyzed the association of pedestrian evacuation with the fire dynamics, pedestrian densities, and fire positions. Several studies also performed the pedestrian movement research under the real conditions rather than the simulated condition. Porzycki et al. [7] explored the influence factors of pedestrian evacuation risk under real evacuation experiments (i.e., the artificial, non-toxic smoke scenario of road tunnel). The results showed that the pedestrian evacuation was influenced by evacuees’ attitudes, familiarity with environment, and the visibility level. Rio and Warren [8] studied the real-world crowds walking behavior using naturalistic data. The results provided an empirical basis for selecting, calibrating, and validating models of the local interactions between pedestrians.

In addition to the influence factor analysis above, some literatures have studied pedestrian behaviors of subway station such as queuing behavior and route-choice behavior. The studies about the queuing behavior of pedestrians in subway stations can be classified into two categories: macroscopic analytical models and microscopic simulation models. Macroscopic analytical models were generally based on queue theory [9,10,11]. Microscopic simulation models aimed to describe pedestrian’s movement behaviors, and to explore the space-time evolution of pedestrian flows. Several scholars have established the microscopic simulation models to analyze the pedestrian behavior in subway station [12,13,14,15]. 

With respect to route-choice behavior, Stubenschrott et al. [16] stated that the pedestrian route-choice behavior in the evacuation process should consider two concepts: the familiarity to the infrastructures, and the internal preference for selecting a certain path. Some studies on pedestrian route-choice behaviors under different conditions have been conducted [16,17,18]. 

Aside from the above studies about the influence factor analysis of pedestrian evacuation, and the pedestrian behaviors research of subway station. Some studies also evaluated the pedestrian evacuation risk of subway station. Yan et al. [19] evaluated the crowd crushing and trampling risk in subway stations based on data envelopment analysis. Then, the area of accident-prone place and accident cumulative duration were selected as the pedestrian risk evaluation indicators. Wang [20] established a pedestrian crowded stampede risk model of subway station based on empowering related degree method. The number of accidents and number of casualties were used to assess the pedestrian risk. 

Overall, previous studies were primarily studied under the regular subway operating period, whereas the research studies under large-scale sports activity scenario are limited. During large-scale sport activity, the subway station operating is more complicated, and the pedestrian flow is higher than that of the regular period. In the meantime, the pedestrian movement presents some spatial aggregation characteristic. Under these circumstances, the pedestrian risk around the subway station is larger than that of the regular period, and the emergency events related to pedestrians are more likely to be triggered. Therefore, the pedestrian risk evaluation of subway station under the large-scale sport activity scenario should attract more research concern, and the need for pedestrian risk assessment is also larger than that under the regular scenarios. In addition, previous studies only used a single risk indicator, or several separate risk indicators to assess the pedestrian risk. Few studies have considered the pedestrian evacuation risk assessment based on multi-indicators. Actually, the subway transportation system is a rather complex system; the factors that influence the subway operation and pedestrian evacuation are multitudinous. So, using a single indicator to evaluate the pedestrian risk is too idealistic. The current study fills these gaps, as the research scenario in this study is large-scale sport activity. In addition, a risk assessment model integrates multi-indicators (i.e., the pedestrian stampede probability and pedestrian casualty) is proposed. The proposed model is more reliable compared to previous model that uses only one indicator, because it not only includes the likelihood of the pedestrian risk (i.e., pedestrian stampede probability) but also takes the severity of the risk (i.e., pedestrian casualty) into consideration. The research results have a positive supporting for the pedestrian risk management at subway stations in large-scale sport activity scenario, and the management flowchart shown as Figure 1.

## 2. Method

### 2.1. Study Corridor 

The study corridor is the “Olympic Park Station” of Beijing, China. It is an important subway station of Beijing metro line 8 that connects many major sports stadiums. The subway station is close to Olympic Village where many important sport venues such as the Bird’s Nest Stadium, National Stadium of China, and National Swimming Center of China are located. During the 2008 Beijing Olympic games, this station took on the main task of transporting tourists. 

The studied subway station is a two-story underground station with island platform design (see Figure 2a); the lower ground is the station platforms where the pedestrians get on and off the subway, and three escalators are set between these station platforms. Then, the upper ground transports the passengers from the lower ground, and it is also connected with the exit channel. The whole station length is 200 m, and the station platform width is 25 m, then five exits (i.e., I, B, C, D, E) are open, and the channel lengths of the five exits are described in Figure 2b. 

### 2.2. Social Force Model

Generally, as the components of pedestrian evacuation analysis, the pedestrian modeling is essential. Two major pedestrian modeling methods are commonly used: the cellular automata model (CAM) and the social force model (SFM). Considering the CAM is limited in the range of speed and direction changing, the SFM is selected as the pedestrian evacuation model in this study. 

SFM is a continuous model that preferably describes pedestrian microscopic movements, and analyzes the relationship between individual and group behavior. SFM was first proposed by Helbing and Molnar [21]. Subsequently, SFM is widely used for simulating and analyzing the pedestrian movement characteristics [22,23]. The desire force, repulsive force, and attractive force, are contained in the SFM. The equations of SFM are shown below:(1)$$m_{\alpha }=\frac{d\vec{\omega_{\alpha }}}{dt}=\vec{F_{\alpha }\left(t\right)}+\xi$$
(2)$$\vec{F_{\alpha }\left(t\right)}=\vec{F_{\alpha }^{0}}\left(\vec{v_{\alpha }},v_{\alpha }^{0}\vec{e_{\alpha }}\right)+\underset{\beta }{\sum }\vec{F_{\alpha \beta }}\left(\vec{e_{\alpha }},\vec{r_{\alpha }}-\vec{r_{\beta }}\right)+\underset{B}{\sum }\vec{F_{\alpha B}}\left(\vec{e_{\alpha }},\vec{r_{\alpha }}-\vec{r_{B}}\right)+\underset{i}{\sum }\vec{F_{\alpha i}}\left(\vec{e_{\alpha }},\vec{r_{\alpha }}-\vec{r_{i}},t\right)$$
where $m_{\alpha }$ is the particle mass, $\vec{\omega_{\alpha }}$ is the desired velocity, $\vec{F_{\alpha }\left(t\right)}$ is the magnitude force. Terms on the right of Equation (2) contain the desired force, repulsive force and attractive force. ξ means a fluctuation term that represents random variations of the behavior.

Desire force determines the force that drives people to their destinations, which is expressed as Equation (3): (3)$$\vec{F_{\alpha }^{0}}\left(\vec{v_{\alpha }},v_{\alpha }^{0}\vec{e_{\alpha }}\right)=m_{\alpha }\frac{1}{\tau_{\alpha }}\left(v_{\alpha }^{0}\vec{e_{\alpha }}-\vec{v_{\alpha }}\right)$$

If not disturbed, a pedestrian will walk into the desired direction $\vec{e_{\alpha }}$ with certainly desired speed $v_{\alpha }^{0}$. A deviation between the actual velocity $v_{\alpha }^{0}\vec{e_{\alpha }}$ due to fluctuation can be modified by relaxation time $\tau_{\alpha }$.

Repulsive force contains repulsive forces among pedestrians, and between pedestrians and obstacles. The first part is expressed as:(4)$$\vec{f_{\alpha \beta }^{soc}}\left(t\right)=A_{\alpha }exp(\frac{r_{\alpha \beta }-d_{\alpha \beta }}{B_{\beta }})\vec{n_{\alpha \beta }}$$
(5)$$\vec{f_{\alpha \beta }^{ph}}\left(t\right)=k\Theta \left(r_{\alpha \beta }-d_{\alpha \beta }\right)\vec{n_{\alpha \beta }}+k\Theta \left(r_{\alpha \beta }-d_{\alpha \beta }\right)\Delta v_{\alpha \beta }^{t}\vec{t_{\alpha \beta }}$$

Suppose α is pressed by β. $r_{\alpha \beta }$ is the sum of the two persons’ radius. $d_{\alpha \beta }$ denotes the distance between two pedestrians’ centers of mass. $\vec{n_{\alpha \beta }}$ is the normalized vector. $\vec{t_{\alpha \beta }}$ is the tangential direction.

The second part is primarily the force between pedestrians and borders of walls and obstacles, which expressed as Equation (6), where *B* means the border.
(6)$$\vec{f_{\alpha B}}\left(t\right)=A_{\alpha }exp(\frac{r_{\alpha B}-d_{\alpha B}}{B_{\alpha }})\vec{n_{\alpha B}}+k\Theta \left(r_{\alpha }-d_{\alpha B}\right)\vec{n_{\alpha B}}+k\Theta \left(r_{\alpha }-d_{\alpha B}\right)\left(v_{\alpha ,}\vec{t_{\alpha B}}\right)\vec{t_{\alpha B}}$$

Pedestrians will be attracted by other people or objects when they are walking. The attractive force between two persons is expressed as Equation (7), where $n_{\alpha \beta }$ is a constant.
(7)$$\vec{f_{\alpha \beta }^{att}}=-C_{\alpha \beta }n_{\alpha \beta }\left(t\right)$$

Many works have analyzed the pedestrian evacuation process using SFM under different environments such as airports [24], shopping malls [25], as well as metro stations [26]. In this study, we aim to analyze the pedestrian evacuation process of subway station, and evaluate the pedestrian evacuation risk in large-scale sport activity scenario.

### 2.3. Simulation Design

Under regular circumstances, the daily pedestrian flow of the “Olympic Park Station” is less than 100,000 people, while during large-scale sport activity, the pedestrian flow of this station grows dramatically. Relevant statistics showed during the large-scale sport activity (taking 2008 Beijing Olympic Games as an example), the daily passenger turnover increases greatly, up to more than 150,000 people. In this condition, the pedestrian evacuation is difficult and the pedestrian stampede risk increases. Therefore, in order to present a more detailed analysis about the pedestrian evacuation of the subway station, the pedestrian evacuation simulations under two different scenarios (i.e., the regular subway operation scenario and large-scale sport activity subway operation scenario) are conducted, respectively. These simulations aim to assess the pedestrian density and related pedestrian risk in the evacuation process. 

In large-scale sport activity scenario, the pedestrian flow in subway station increases sharply in a short time, which may lead to congestions at the subway exits or at escalators. In this condition, if the security measures of the subway station are not timely and unscientific, the pedestrian stampede event prone to happen. Many studies have proved that the pedestrian stampede is very easy to occur under the condition of the mass pedestrian gathering. 

Thereby, we attempt to simulate the pedestrian movement of subway station under large-scale sport activity scenario, which is helpful to understand the pedestrian motion characteristics and analyze the pedestrian risk. The simulation setting shown as Figure 3a, where the exits from left to right are D/E exits, C exit, and B/I exit. Then, three escalators are included in the subway station, and two groups of ticket gates are set at the left and right side of the station. Figure 3b shows the pedestrian movements in the evacuation process

We assume four typical positions where the pedestrian stampede event may occur during large-scale sports activity. The first position locates at between the left ticket gate and the left escalator (i.e., location 1). The mass of passengers from the left escalator will pass the left ticket gate, especially when the pedestrian passing speed is slower than that transported by the escalator, large amounts of pedestrians will gather near the ticket gate. In this case, the pedestrian surge phenomenon will appear, which will induce pedestrian stampede. Location 4 shows the same situation with location 1. Then location 2 locates at between the left escalator and the middle escalator. As the nearest exit to location 2 is C exit, most of the pedestrians between the left and middle escalator will select C exit as the evacuation route in emergency condition. In this process, the pedestrian movement from different directions at location 2 may produce crossover, which boosts the pedestrian congestion and increases the pedestrian stampede risk. Likewise, the same phenomenon may be presented at location 3.

The pedestrian movement rules in the simulation are set as follows. We first assume that one subway shuttle of single direction arrives at certain time, then the pedestrians get off the subway from one side, and they will select the expected escalator to evacuate. For the left escalator, we presume that most of the pedestrians will choose D/E exit to evacuate, and this probability is set as 0.7. Then, the probability to choose C exit is 0.2, and to choose I /B exit is 0.1. For the middle escalator, the probability to choose D/E exit, C exit, and I/B exit are equal, and they are all 0.33. As for the right escalator, the probability to choose I/B exit is 0.7, to choose C exit is 0.2, and to choose D/E exit is 0.1. Assuming the subway comes every 5 minutes, and working 12 hours a day (i.e., from 8:00 am to 8:00 pm). Then the number of pedestrians taking each of the escalator is expressed as follows.
(8)$$n={\frac{N_{1day}}{12\times 12\times 3}}_{5min}$$
where $n_{5min}$ is the pedestrians taking each of the escalators every five minute, $N_{1day}$ denotes the daily pedestrian flow from 8:00 am to 8:00 pm. 

### 2.4. Pedestrian Evacuation Risk Assessment Model

Classical risk assessment theory regards the risk rate as an effective indicator to measure the risk value [27], in which the risk rate is expressed by the product of the risk probability (*P*) and risk severity (*S*). Based on the classical risk assessment theory, this study proposes a quantitative pedestrian evacuation risk assessment model that combines the pedestrian stampede probability (i.e., the pedestrian stampede probability under large-scale sport activity) with the severity of this stampede event (i.e., the pedestrian casualties), as expressed as:(9)$$R_{p}=\theta \times P_{pr}\times S_{pc}$$
(10)$$P_{pr}=\frac{N_{i}}{N}$$
where $R_{p}$ is pedestrian evacuation risk under large-scale sport activity, $P_{pr}$ is stampede probability, $S_{pc}$ represents the consequence severity of the stampede event. θ is the trigger factor of pedestrian stampede, and it represents the correlation coefficient between the pedestrian stampede likelihood and the caused severity. The trigger factor contains three aspects: proportion of vulnerable population (i.e., the elderly, children, and disabled), exit attraction (i.e., the familiarity to the exit location), and crowd information (the communication between pedestrians, and between pedestrians and subway station managers). The objective function of the trigger factor is complex, containing the detail information about the population composition, exit design of the subway station, and the specific management and scheduling. For the convenience of subsequent study and simulation analysis, the trigger factor is assumed to be a fixed value (it is set as 0.5). 

Then the pedestrian stampede probability is expressed by the ratio of the stayed pedestrians (i.e., $N_{i}$, which denotes that no evacuated pedestrians at time *i*) to total pedestrians (i.e., N) in the station. Notably, the pedestrian casualty (i.e., $S_{pc}$) is defined as the pedestrian who fails to be evacuated within the stampede event influence area when the evacuation ends, which aligns with the default setting of “AnyLogic”.

## 3. Result and Analysis

### 3.1. Random Evacuation Simulation 

The first 600 s in the simulation is the warm up time. After 600 s, the pedestrian evacuation starts. The simulation duration is set as 300 s, which aligns with the evacuation standards of subway stations in China. In order to evaluate the pedestrian evacuation risk under different scenarios, two simulation corresponding to regular subway operating scenario, and the large-scale sport activity scenario are conducted. Under the two scenarios, the pedestrian flow is also different. Based on previous experience (i.e., 2008 Beijing Olympic Games), the pedestrians taking each of the escalators every five minutes under the two different scenario is 231 people and 347 people, respectively.

Notably, the lower ground mainly simulates the process of pedestrians getting off the subway and taking the escalators. Then, the upper ground simulates the evacuation process of pedestrians after they taking the escalators. The pedestrian densities of both the lower and upper ground can be obtained in the simulation. But the pedestrian trample probability and the related pedestrian risks are primarily those occurred at the upper ground in our design. Because only the pedestrian groups have arrived in the upper ground, the evacuation can be implemented, and this process is the key part of the simulation. Finally, the changes of pedestrian density, trample probability, and pedestrian risk in the evacuation process are shown in Figure 4.

At the beginning, the pedestrian density of the upper ground for both the regular and large-scale sport activity scenarios are higher than that of the lower ground. As time goes on, the pedestrians in the upper ground are gradually evacuated. Then, the pedestrians in the lower ground continue to increase, and the density keep increase. At a critical time, the pedestrian density of the upper ground is equal the lower ground. For the regular scenario, the critical time is t_1_ = 700 s. Then the critical time in the large-scale sport activity scenario is t_2_ = 710 s. Afterwards, the pedestrian density of the upper ground is decreasing, and it is lower than that of the lower ground, until the evacuation ends. In addition, the pedestrian densities of the large-scale sport activity are always larger than that of the regular scenario both on the lower and upper ground. This reflects that the scheduling frequency of subway in large-scale sport activity is higher than that of the regular condition. Due to this situation, the pedestrian stampede likelihood in large-scale sport activity also increase (see Figure 4b), leading to a higher pedestrian evacuation risk (see Figure 4c). 

### 3.2. Shortest Path Evacuation Simulation

The above processes have analyzed the pedestrian evacuation characteristics from both the regular scenario and large-scale sport activity scenario, but these processes simulate the random evacuation of pedestrians. However, in real condition, some ordered evacuation may achieve a better effect, which may reduce the pedestrian evacuation risk. So, apart from the random evacuation conducted above, we also design a shortest path evacuation scheme to simulate the pedestrian movement of subway station in large-scale sport activity. 

In the shortest path evacuation, the pedestrians will choose the nearest exit to evacuate instead of selecting the random exit. Thereby the pedestrian movements will be more ordered in this evacuation process, and the average pedestrian stampede likelihood is also lower. Figure 5 presents pedestrian trample probability of the two evacuation ways under large-scale sport activity. At the beginning (i.e., 600 s–750 s), the pedestrian stampede likelihoods of the two evacuation ways show the similar trends. As the pedestrian density at this period is low, and the pedestrian groups by different escalators need time to search for their shortest paths. Afterwards (i.e., after 750 s), the pedestrian stampede likelihoods of the two evacuation ways become to present obvious difference. Because during this process, the pedestrians by the shortest path evacuation have found their evacuation paths and would move towards them. Then the pedestrians by the random evacuation still moves with a disorganized way. As a result, the pedestrian trample likelihood of the shortest path evacuation declines more than that of the random evacuation (see Figure 5). 

Then the average pedestrian risk of the shortest path evacuation is also lower than that of the random evacuation (see Figure 6). Specifically, the pedestrian risk of the shortest path evacuation under large-scale sport activity is an increasing and decreasing process. First, at *t* = 600 s, the pedestrian density in the upper ground has achieved a high level, as large numbers of people have been transported to the upper ground at this time period. If the pedestrian stampede occurs at this time period (we first assume the stampede occurred at location 1), the pedestrian emergency evacuation starts. At this period, the pedestrian density in the upper ground is higher than lower ground. As times go, some pedestrians in the upper ground have been evacuated successfully, then the pedestrian density and pedestrian evacuation risk decreases. When *t* = 710 s, the pedestrian density in the upper ground is equal to that of the lower ground, then the mass pedestrians in lower ground begin to take the escalator urgently, and start to search for the nearest exit. During this process, the superposition, crossing and detour distance between pedestrians of different directions increase the evacuation risk, which will last until *t* = 780 s (see Figure 6). After 780 s, most of the pedestrians have found their shortest paths, and begin to evacuate along the shortest path exits. In this condition, the evacuation presents a good order, and the pedestrian risk decreases again until the evacuation ends. 

### 3.3. Different Simulation Scenarios Using Shortest Path Evacuation 

To further assess the impact of different stampede locations on pedestrian evacuation risk, we simulate the pedestrian evacuation of different stampede locations using the shortest path evacuation (see Figure 7a). Overall, when the pedestrian stampede position is location 1 or location 4, the pedestrian evacuation risk is relatively low. By contrast, it is high when the pedestrian stampede position is location 2 or location 3. This illustrates that when the stampede event locates at the edge of the station (i.e., location 1, location 4), the pedestrian evacuation is relatively easy, and the involved pedestrian risk is also low. On the contrary, when the pedestrian stampede located in the middle part of the subway station (i.e., location 2, location 3), the evacuation become difficult, and the corresponding pedestrian risk is also high. 

Because when the stampede event happens in the middle part of the station, the evacuation exits within the event’s influence area will be covered with a high probability. For example, if location 2 occurs a stampede event, the E /D exit, and C exit will be influenced (see Figure 3), because most of the pedestrians taking the left and middle escalator will choose these exits to evacuate. In this case, the pedestrian density increases, leading to a difficulty evacuation. So, the pedestrian evacuation risk also increases. Likewise, these characteristics also appear when the stampede event occurs at location 3. However, when the stampede position is location 1 or location 4, the influence area only covers E/D exit, or B/I exit, so the average evacuation risk is obviously low. 

A further analysis shows the average pedestrian risk under the condition of “stampede occurs at location 1” is lower than that of “stampede occurs at location 4”. This is attributed to that the distance from location 1 to the left ticket gate is longer than the distance from location 4 to the right ticket gate, thus the pedestrian evacuation is relatively convenient. With respect to the situation of “stampede occurs at location 1, or location 2”, the influence area will cover E exit, D exit, C exit (for location 2), and C exit, B exit, I exit (for location 3) respectively. The evacuation exits on the both edges under the two different scenarios may be influenced with the same likelihood. But the stampede event under the two different locations have different impacts on C exit. 

Specifically, location 2 is closer to C exit, so once this position occurs a stampede event, the pedestrians will get together in a short time, leading to a high risk. While location 3 keeps a long distance from C exit, so the pedestrians have enough space to move to C exit once a stampede event occurs. Under this circumstance, the pedestrian evacuation risk is relatively low. That’s why the curve of location 2 (see Figure 7a) is in the above of the curve of location 3 at the beginning. After 710 s, the curve of location 2 is below the curve of location 3, which indicates the pedestrians nearby location 2 have adapted to the emergency environment, and they are evacuated through C exit quickly. However, under the equal condition, the pedestrian evacuation risk of location 3 is larger than that of location 2 after 710 s, which demonstrates the pedestrians in location 3 need time to evacuate through C exit. 

The above analyses are only available under the condition of “pedestrians get off from one side”. To explore the relationship between the “get off” way and the pedestrian evacuation risk, we simulate the pedestrian evacuations of different “get off” ways (i.e., get off from one side, and get off from both sides). “Get off from one side” indicates that the subway arrives in from one direction, and “get off on both side” means that the subway arrives in from two directions at the same time. In this simulation, other variables are constant (i.e., assuming the stampede event is location 1, the evacuation way is the shortest path evacuation, and the scenarios are the large-scale sport activity), then the simulation result shown as Figure 7b. Before 710 s, the pedestrian evacuation risk under two different conditions are the same. After 710 s, the pedestrian evacuation risk of “get off from one side” is slightly higher than that of “get off from both sides”. But the risk difference is small, indicating that the “get off” way presents no obviously impact on pedestrian evacuation risk.

### 3.4. Pedestrian Evacuation Risk Analyses from Temporal Perspective 

From the analyses of pedestrian movement characteristics and pedestrian evacuation risk above. It is found that the factors highly associated with the pedestrian evacuation risk of the studied subway station are “whether it involves a large-scale sport activity”, the “pedestrian evacuation method”, and “pedestrian stampede event location”. Then the “get off” way has a small influence on pedestrian evacuation risk. 

After determining these factors, we further analyze the pedestrian density at different evacuation time stamps, and make a further assessment of the pedestrian evacuation risk over evacuation time. At first, at *t* = 600 s, the pedestrians begin to gather and cause congestions. In this case, the pedestrian density is high, leading to a high risk (see Figure 8a). After 600 s, the evacuation starts, the mass pedestrians in the upper ground begin to be evacuated. The pedestrians nearby the exits are first to be evacuated, then the pedestrian density and pedestrian evacuation risk decrease in this process (see Figure 8b). As times go on, the pedestrians far from the exits start to search for the shortest evacuation paths. In this process, the pedestrian distribution is extremely uneven, as some local areas such as the C exit channel may be selected as the same evacuation path, so the pedestrian density increases, as shown in Figure 8c. As a result, the pedestrian crossover and overlap from different directions further enhance the evacuation risk. Afterwards, the pedestrians are evacuated persistently from the shortest routes, and the pedestrian density reduces until the whole evacuation ends. So, the pedestrian density and the pedestrian evacuation risk also decrease (see Figure 8d). 

The whole process above estimates the changes of pedestrian density and pedestrian evacuation risk over time. Combining the results of this temporal analysis with the pedestrian evacuation influence factors analyses above, the subway organizers can make some target measures to prevent the pedestrian stampede of subway stations under large-scale sport activity, and reduce the pedestrian risk.

## 4. Discussion

Based on SFM and simulations, we present a quantitative evaluation method of pedestrian evacuation risk of subway station in large-scale sport activity scenario. Compared with previous studies about the pedestrian behavior evaluation in subway station, this study proposes a more reliable pedestrian evacuation risk evaluation model that considers not only the likelihood of the pedestrian risk (i.e., using the pedestrian stampede probability as the assessment indicator) but also the severity of the risk (i.e., pedestrian casualty is used as the assessment indicator). Then several different scenarios in large-scale sport activity are simulated, and the pedestrian evacuation risks and their corresponding influence factors are analyzed.

The results demonstrate that the pedestrian evacuation methods, and pedestrian stampede event locations have a large impact on pedestrian evacuation risk of subway station in large-scale sport activity. Specifically, the shortest path evacuation is a more reasonable evacuation method compared with the random evacuation in large-scale sport activity. The stampede event locates at the edge of the station platform (i.e., location 1 and location 4) show a smaller pedestrian evacuation risk than that of the middle part of the subway station platform (i.e., location 2 and location 3) during the evacuation process. By further contrast, the pedestrian evacuation risk under the situation of “stampede event position is location 1” is lower than that of “stampede event position is location 4”. As location 1 is far from the left ticket gate, and location 4 is close to the right ticket gate. Thus, the scenario of “stampede event position is location 1” has more space to buffer the large number of pedestrians. Finally, the average pedestrian risk of the “get off from both sides” is slightly lower than that of the “get off from one side”.

After identifying the factors associated with pedestrian evacuation risk, the safest scenario for the “Olympic Park Station” in large-scale sport activity is determined (i.e., the evacuation mode is the shortest path evacuation, and the stampede event location is location 1). Then the pedestrian evacuation risk and pedestrian density over time are also analyzed. This process further proves the complex changes of the pedestrian evacuation risk in large-scale sport activity. All of these analyses have proved that the proposed pedestrian evacuation risk assessment model is feasible.

## 5. Conclusions

The proposed study has the dual role of theoretical research and engineering application. First, the developed pedestrian evacuation risk assessment model provides a proactive method to quantify the pedestrian related risk of subway station under large-scale sport activity scenario. Because there has almost no studies attempt to quantify the pedestrian evacuation risk under the large-scale sport activity scenario. Thus, the developed pedestrian evacuation risk quantitative model that takes the double evaluation indicators such as the pedestrian stampede probability and pedestrian casualty into account will lays a theoretical foundation for the future research. In addition, the research results can also be used in the practical engineering application. The results can be referred for the risk identification that involves in pedestrian evacuation of the subway stations under large-scale sport activity, which ensures the safety of the subway operation and reduces the pedestrian related risks. The application of these results also supports the safe and smooth conduction of large-scale sport activity. For example, the subway organizers can implement pedestrian controlling measures before the stampede event at the key locations such as the middle part of the “Olympic Park Station”. They can also limit the pedestrian flow that entering the subway platform by adjusting the get off ways at different time period, which prevents the pedestrian stampede event. Then the effective evacuation method such as the shortest path evacuation should be taken to evacuate the mass pedestrians. Overall, all the analyses results can be used for supporting the decision and management that involves subway operation and scheduling, pedestrian evacuation, and emergency prevention (see Figure 1). 

However, before these results are effectively applied, several efforts are still needed. First, the daily pedestrian flow of the “Olympic Park Station” refers previous data, and the real-time pedestrian flows need to be provided when analyzing the pedestrian evacuation risk under large-scale sport activity. In addition, the stampede event locations and other simulated settings such as the subway platform size, escalator load, etc., are all the hypothetic condition rather than the real situation, so the detailed and real-world information about them may be more applicable for evaluating the pedestrian evacuation risk of subway stations under large-scale sport activity. Thus, the authors suggest that future researches should focus on these issues. 

## Figures and Tables

**Figure 1 ijerph-17-03844-f001:**
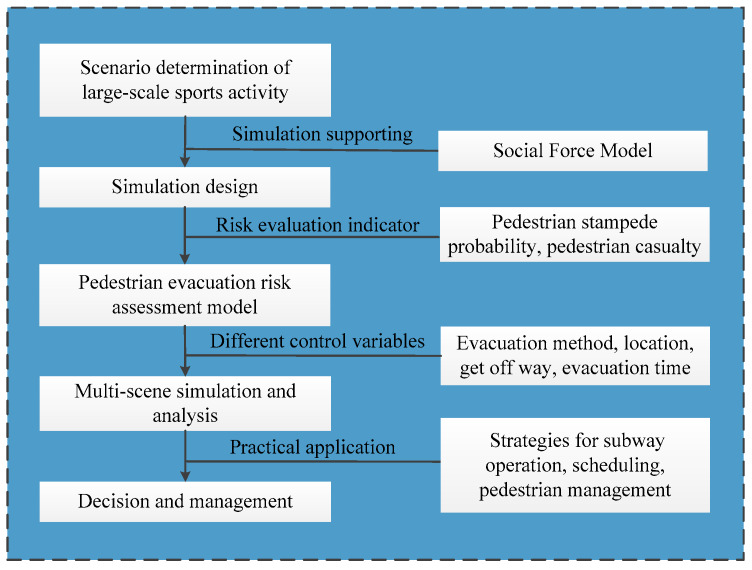
Pedestrian risk management flowchart.

**Figure 2 ijerph-17-03844-f002:**
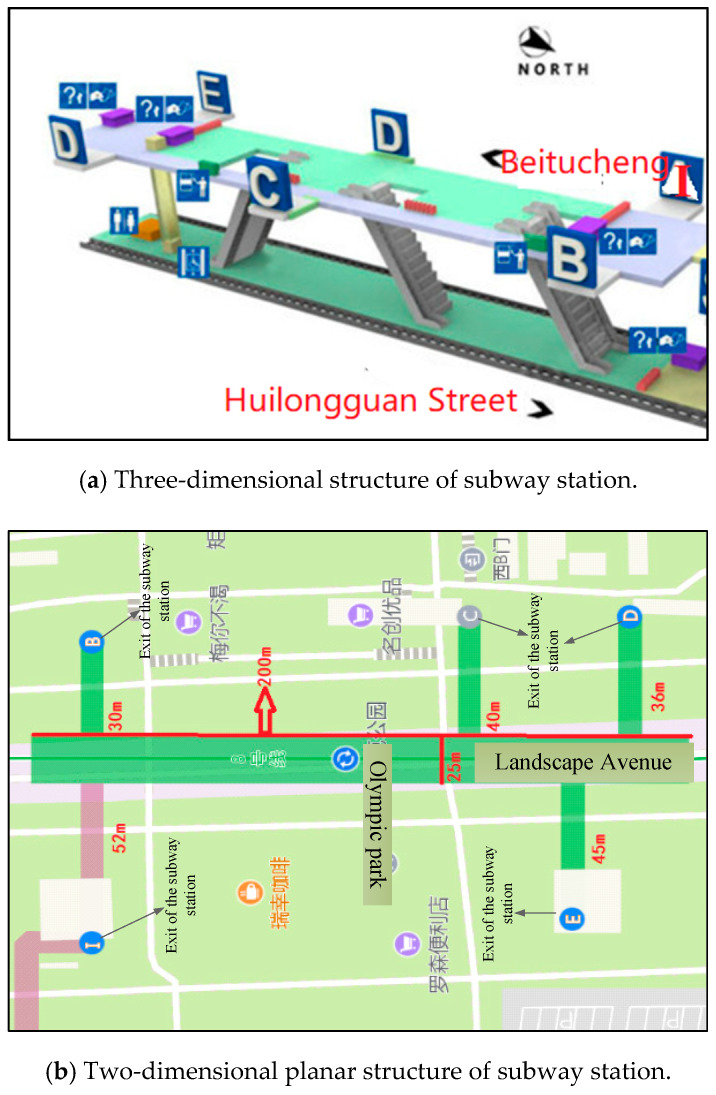
Layout of “Olympic Park Station”.

**Figure 3 ijerph-17-03844-f003:**
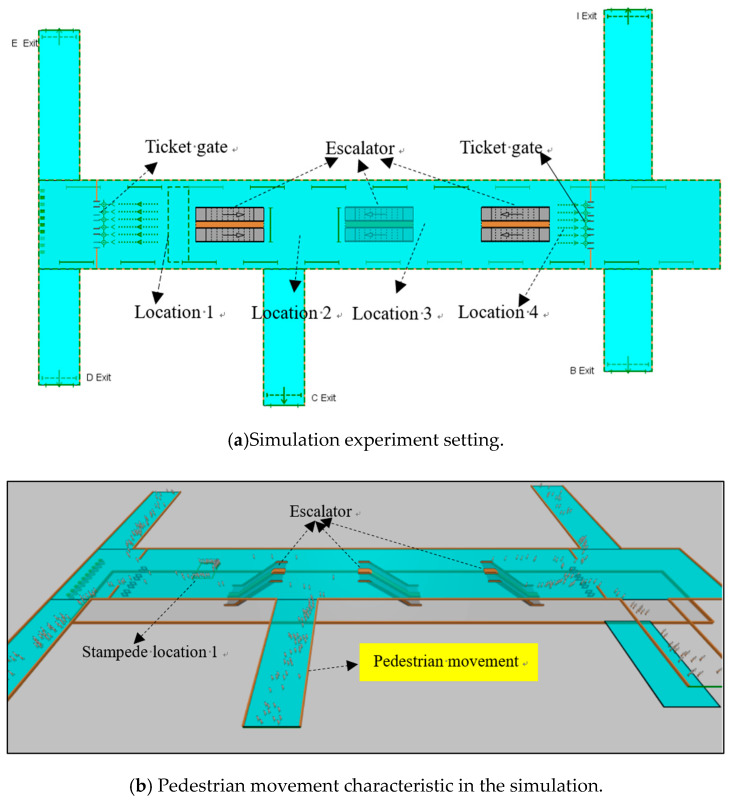
Simulation setting and pedestrian movement of the subway station.

**Figure 4 ijerph-17-03844-f004:**
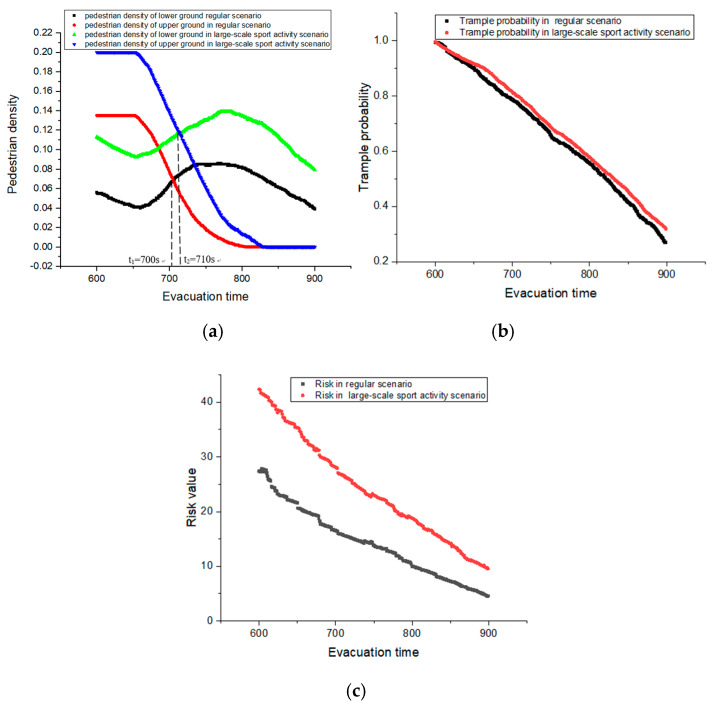
Random evacuation simulation: (**a**) pedestrian density changes, (**b**) trample probability changes, and (**c**) risk changes.

**Figure 5 ijerph-17-03844-f005:**
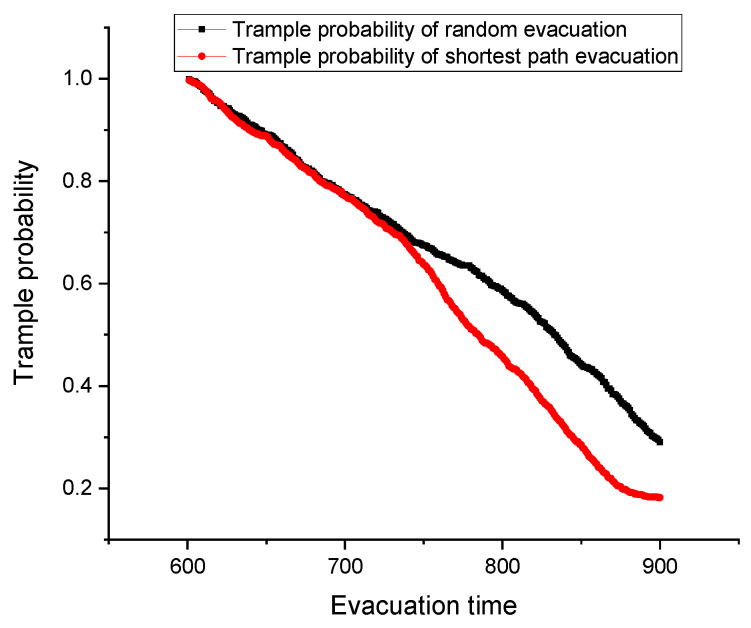
Pedestrian trample probability changes under two evacuation methods.

**Figure 6 ijerph-17-03844-f006:**
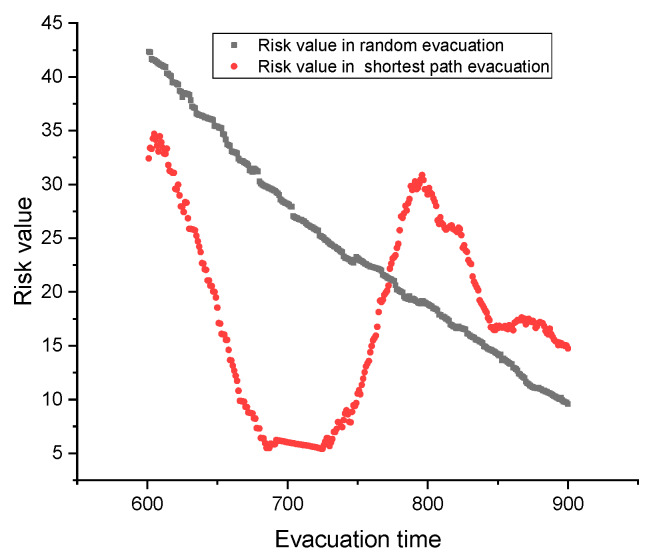
Pedestrian risks under two evacuation methods.

**Figure 7 ijerph-17-03844-f007:**
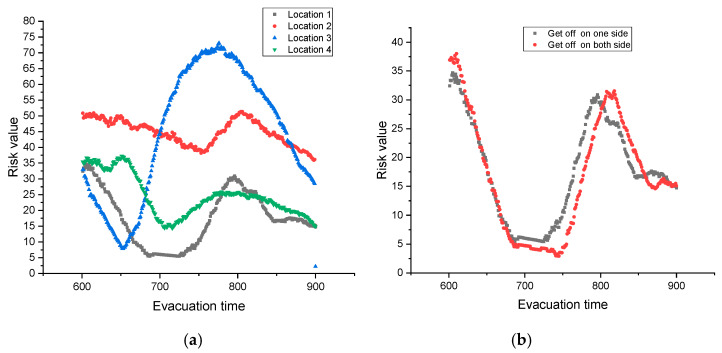
Pedestrian evacuation risk under different scenarios: (**a**) different stampede locations, (**b**) different “get off” ways.

**Figure 8 ijerph-17-03844-f008:**
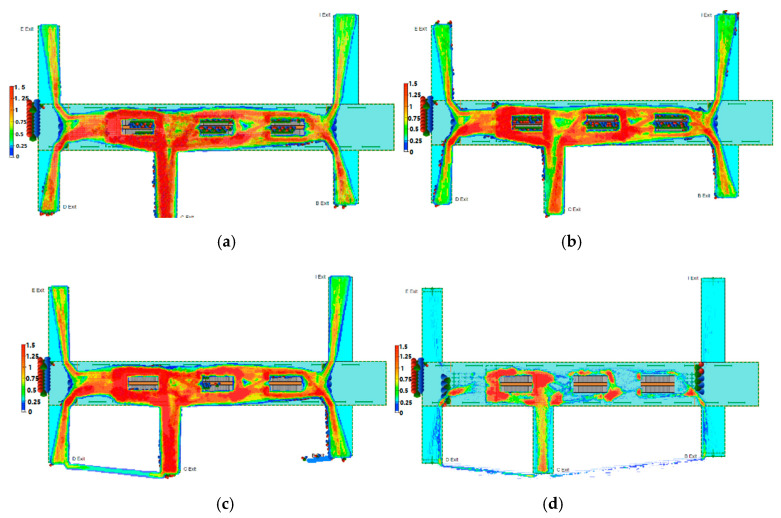
Pedestrian heatmap at different time stamps: (**a**) *t* = 600 s, (**b**) *t* = 700 s, (**c**) *t* = 800 s, (**d**) *t* = 900 s.

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
