# Peer review of "Pedestrian Evacuation Risk Assessment of Subway Station under Large-Scale Sport Activity"

_ijerph, 2020, doi:10.3390/ijerph17113844_

Round 1

Reviewer 1 Report

This paper proposes a method for risk assessment of pedestrian evacuation of subway

stations. The language of the paper is good, and its structure is well organized. Although it is an interesting approach, there are some issues that the authors have to address:

  1. The abstract must be reorganized to show the contribution of the present study.
  2. The authors should add a flowchart illustrating the how the process is applied to the problem for risk assessment
  3. The authors must explain if there are alternative methods for risk assessment in that field and why the proposed approach is better preferred.
  4. The conclusions section must be extended in order to include more information about the contribution of this study to the body of knowledge

Author Response

Dear professor,

Thank you very much for your comments on our manuscript. Based on your suggestions, we have carefully revised the manuscript. Please see our point to point responses to all your comments below. We look forward to hearing from you soon for a favorable decision.

The following are our point to point responses to your comments.

 Comment 1: The abstract must be reorganized to show the contribution of the present study.

Responses: As suggested by you, we have reorganized the abstract, the new version can highlight the contribution of the present study, see the adding part in the revised version.

 Comment 2: The authors should add a flowchart illustrating the how the process is applied to the problem for risk assessment

Responses: As suggested by you, we have added a flowchart illustrating the application of the risk assessment, see Fig. 1 in the revised version.

Figure 1. Pedestrian risk management flowchart

Comment 3: The authors must explain if there are alternative methods for risk assessment in that field and why the proposed approach is better preferred

Responses: Based on your recommendation, we have explained other alternative methods in the introduction section (see the reference [7], [8]). In addition, we have also explained why the proposed approach is better preferred, see the revision in the last paragraph in introduction section.

Comment 4: The conclusions section must be extended in order to include more information about the contribution of this study to the body of knowledge

Responses: As suggested by you, we have extended the conclusion section, and the new version will include the major contribution (include the contribution of theoretical research and engineering application) of this study, see the first paragraph and the second paragraph of the discussion and conclusion section with the "Track Changes" in the revised version.

All the above are the responses to your comments.

Wish you all the best

Yours sincerely,

Zeyang Cheng, Jian Lu, and Yi Zhao

Reviewer 2 Report

Well done manuscript.

Only minor spell check required.

Author Response

Dear professor,

We are truly grateful to your kindly comments. Based on your comments, we have carefully checked the spell problem and revised them, see the "Track Changes" in the new version.

Wish you all the best!

Yours sincerely,

Zeyang Cheng, Jian Lu, and Yi Zhao

Reviewer 3 Report

This is a very interesting study providing a risk assessment of pedestrian evacuation in subway stations after a large event. The paper is well-written and the methodology employed seems appropriate; the findings also look credible. I have some comments intended to clarify some parts of the analysis:

  1. Even though the analysis makes sense, the contributions of the study to the existing state of knowledge need to be demonstrated in more detail. If, for example, the combination of various indices (pedestrian stampede probability and pedestrian casualty) for the risk assessment is an advancement compared to previous approaches, this should be explicitly stated mentioning also how the use of multiple indicators can benefit the analysis.
  2. The literature review is overall adequate. The authors could perhaps consider a brief commentary on studies using naturalistic data (apart from simulation) for pedestrian evacuation or any other pedestrian-involved movement, as an alternative methodological approach:

https://www.sciencedirect.com/science/article/pii/S2213665718300253

https://www.ncbi.nlm.nih.gov/pmc/articles/PMC6114522/

https://www.sciencedirect.com/science/article/pii/S2352146514000532

  1. Several minor typos have been observed throughout the text. The authors are strongly encouraged to carefully proofread the manuscript, perhaps with the help of an editing service or a native speaker.

Author Response

Dear professor,

Thank you very much for your constructive comments and valuable suggestions. Based on your suggestions, we carefully revised the manuscript. Please see our point to point responses to all your comments below.

Comment 1: Even though the analysis makes sense, the contributions of the study to the existing state of knowledge need to be demonstrated in more detail. If, for example, the combination of various indices (pedestrian stampede probability and pedestrian casualty) for the risk assessment is an advancement compared to previous approaches, this should be explicitly stated mentioning also how the use of multiple indicators can benefit the analysis

Response: Based on your suggestion, we have made a detailed introduction of the contributions to this study, see the revisions in the abstract section, introduction section, and discussion and conclusion section.

Comment 2: The literature review is overall adequate. The authors could perhaps consider a brief commentary on studies using naturalistic data (apart from simulation) for pedestrian evacuation or any other pedestrian-involved movement, as an alternative methodological approach:

https://www.sciencedirect.com/science/article/pii/S2213665718300253

https://www.ncbi.nlm.nih.gov/pmc/articles/PMC6114522/

https://www.sciencedirect.com/science/article/pii/S2352146514000532

Response: As suggested by you, we have added a summarization of the studies using naturalistic data. After checking the links in your comments, we considered that the second and the third studies are related to our research, so we added them as some alternative methodological approach related to pedestrian behavior. See the revision in the second paragraph of Introduction section (literatures [7] and [8]). The two literatures are also listed in the reference section.

Comment 3: Several minor typos have been observed throughout the text. The authors are strongly encouraged to carefully proofread the manuscript, perhaps with the help of an editing service or a native speaker.

Response: As suggested by you, we have carefully checked the typos, and we also ask an English native speaker to help us check the language.

Wish you all the best!

Yours sincerely,

Zeyang Cheng, Jian Lu, and Yi Zhao